# Comprehensive analysis of intercellular communication in the thermogenic adipose niche

Farnaz Shamsi [1,2,3,8 ✉], Rongbin Zheng[4,5,8], Li-Lun Ho[6], Kaifu Chen [4,5 ✉] & Yu-Hua Tseng [1,7 ✉]

Brown adipose tissue (BAT) is responsible for regulating body temperature through adaptive thermogenesis. The ability of thermogenic adipocytes to dissipate chemical energy as heat counteracts weight gain and has gained considerable attention as a strategy against obesity. BAT undergoes major remodeling in a cold environment. This remodeling results from changes in the number and function of brown adipocytes, expanding the network of blood vessels and sympathetic nerves, and changes in the composition and function of immune cells. Such synergistic adaptation requires extensive crosstalk between individual cells in the tissue to coordinate their responses. To understand the mechanisms of intercellular communication in BAT, we apply the CellChat algorithm to single-cell transcriptomic data of mouse BAT. We construct an integrative network of the ligand-receptor interactome in BAT and identify the major signaling inputs and outputs of each cell type. By comparing the ligand-receptor interactions in BAT of mice housed at different environmental temperatures, we show that cold exposure enhances the intercellular interactions among the major cell types in BAT, including adipocytes, adipocyte progenitors, lymphatic and vascular endothelial cells, myelinated and non-myelinated Schwann cells, and immune cells. These interactions are predicted to regulate the remodeling of the extracellular matrix, the inflammatory response, angiogenesis, and neurite growth. Together, our integrative analysis of intercellular communications in BAT and their dynamic regulation in response to housing temperatures provides a new understanding of the mechanisms underlying BAT thermogenesis. The resources presented in this study offer a valuable platform for future investigations of BAT development and thermogenesis.

[1] Section on Integrative Physiology and Metabolism, Joslin Diabetes Center, Harvard Medical School, Boston, MA 02115, USA. [2] Department of Molecular Pathobiology, College of Dentistry, New York University, New York, NY 10010, USA. [3] Department of Cell Biology, Grossman School of Medicine, New York University, New York, NY 10016, USA. [4] Basic and Translational Research Division, Department of Cardiology, Boston Children's Hospital, Boston, MA 02115, USA. [5] Department of Pediatrics, Harvard Medical School, Boston, MA 02115, USA. [6] Computer Science and Artificial Intelligence Laboratory, Massachusetts Institute of Technology, Cambridge, MA, USA. [7] Harvard Stem Cell Institute, Harvard University, Cambridge, MA, USA. [8] These authors contributed equally: Farnaz Shamsi, Rongbin Zheng. ✉email: fs2451@nyu.edu; kaifu.chen@childrens.harvard.edu; yu-hua.tseng@joslin.harvard.edu

Brown adipose tissue (BAT) is a specialized adipose type primarily responsible for regulating body temperature through adaptive thermogenesis. The ability of thermogenic adipocytes to dissipate chemical energy in the form of heat counteracts energy storage and has gained considerable attention as a strategy against obesity. Studies in rodents have demonstrated that increasing the amount or activity of brown adipocytes promotes energy expenditure, improves insulin sensitivity, and protects animals from diet-induced obesity[1–4].

The conventional view of cold-induced thermogenesis involves the sensation of cold by the cutaneous and deep body sensory neurons, which in turn signals to the hypothalamic networks that integrate them with brain temperature information to drive appropriate sympathetic neural outflow to BAT[5]. Acute cold exposure stimulates adaptive thermogenesis by enhancing the thermogenic function of existing brown adipocytes. In contrast, prolonged cold exposure promotes de novo recruitment of brown adipocytes and simultaneous remodeling of other adipose-resident cells to maximize thermogenesis[6]. Specifically, chronic exposure to cold induces a coordinated expansion of brown adipocyte progenitors, endothelial cells, and nerve terminals, as well as changes in the composition of BAT-resident immune cells[7,8]. Although adipocytes are the heat-producing cells in BAT, other cell types form the adipocyte niche and regulate adipocyte number and function through extensive cellular crosstalk.

In recent years, single-cell RNA-sequencing (scRNA-seq) analyses of different adipose depots have provided a comprehensive map of adipose-resident cell populations, including adipocytes and their progenitors, fibroblasts, different types of immune cells, endothelial cells, vascular smooth muscles (VSMs), and Schwann cells[9–12]. By analyzing the cell-type-specific transcriptional changes in BAT of mice housed at different temperatures, we showed that cold exposure instigates significant transcriptional changes in all BAT-resident cells. Many transcriptional changes are accompanied by cell identity transitions, e.g., the differentiation of Trpv1-expressing progenitors to thermogenic adipocytes[9]. However, how distinct cell types in the BAT microenvironment communicate and how the responses of individual cells are integrated to produce a coordinated adaptation have not been described.

Here, we use scRNA-seq data from the interscapular BAT (iBAT) of mice housed at different temperatures to construct the ligand-receptor interactome within the BAT microenvironment. Using CellChat to quantitatively infer intercellular crosstalk from the cell-type-specific gene expression data, we identify the primary signaling inputs and outputs for each cell type in BAT. This analysis reveals the role of adipocyte progenitors as the major communication "hub" in the adipose niche. Notably, comparing the ligand-receptor interactome of BAT in mice housed at different temperatures revealed that cold exposure significantly enhances cellular crosstalk in BAT. Specifically, the number of incoming and outgoing interactions to or from adipocytes, adipocyte progenitors, lymphatic and vascular endothelial cells, myelinated Schwann cells (MSC), non-myelinated Schwann cells (NMSC), and several immune cell types increases in cold. By comparing the expression of ligands and receptors in each cell type, we identify the interactions that are significantly regulated by ambient temperature. Collectively, our integrative analysis of cellular communication in BAT and their dynamic responses to colds provide new insights into the mechanisms of BAT thermogenesis and cold adaptation. This resource will pave the way and serve as a guide for future investigations of BAT development and function.

## Results

### Housing temperature shapes the cellular composition of BAT.
To understand how different cell types in BAT respond to changes in ambient temperature, we isolated the stromal vascular fraction (SVF) of BAT from mice housed at thermoneutrality (TN: 30 °C for 7 days), room temperature (RT: 22 °C) or cold (5 °C for 2 days or 7 days)[9]. We used the total single-cell suspensions and profiled their transcriptome using the 10x Genomics Single-Cell 3' Gene Expression solution. The resulting dataset included 107,679 high-quality cells and 468 million reads. Cells were clustered using a graph-based approach (Scanpy)[13] and visualized using Uniform Manifold Approximation and Projection (UMAP) method[14]. Using a combination of known markers and the specific transcripts expressed in each cluster, we assigned cell-type identities to the clusters (Supplementary Data 1). Overall, we identified 20 distinct clusters representing major cell types in the SVF of mouse BAT (Fig. 1a). We have previously reported the identification and characterization of the non-hematopoietic cells in BAT and their cold-induced adaptations using a fraction of this dataset[9]. In this study, we used the full dataset to comprehensively analyze the effects of housing temperature on cellular composition and intercellular communications in the BAT microenvironment.

Comparing the cellular composition of BAT-SVF from mice housed at different environmental temperatures revealed significant differences at the cell type level between the sample groups (Fig. 1b and Supplementary Fig. 1a–t). Notably, a significantly higher proportion of adipocytes was detected in mice housed in cold for 2 days (Wald test $p$ value 0.0005) (Supplementary Fig. 1a). Given that the SVF isolation step was used to deplete the lipid-laden adipocytes, these remaining adipocytes likely represent the differentiating adipocytes with lower lipid content and buoyancy than the fully mature adipocytes.

In this analysis, we identified multiple immune cell types within the mouse BAT microenvironment. These include B cells, several populations of T cells, such as CD4+ T, CD8+ T, cytotoxic T, and regulatory T (Treg) cells, erythroid-like cells, basophils, neutrophils, natural killer (NK) cells, macrophages, and type 2 innate lymphoid cells (ILC2) (Fig. 1a, b). Interestingly, colder housing temperature increased the proportion of ILC2s (Wald test $p$ value 0.0029) (Supplementary Fig. 1m) and reduced the proportions of cytotoxic T cells (Wald test $p$ value 0.0297) (Supplementary Fig. 1f) and regulatory T cells (Tregs) (Wald test $p$ value 0.0026) in BAT (Supplementary Fig. 1g). ILC2s are the source of Th2 cytokines IL5 and IL13 that promote the recruitment and accumulation of eosinophils and alternatively activated macrophages[15]. The rise in the proportion of ILC2s in BAT of mice housed in cold is consistent with previous work showing that ILC2s directly regulate thermogenic adipocyte development by regulating the number and fate of adipocyte precursors[16, 17]. Previous studies have revealed the contribution of adipose-resident T cells to adipose function and systemic metabolism[18–20]. The proinflammatory cytotoxic T cells were shown to be upregulated in the epididymal white adipose tissue (WAT) of diet-induced obese mice and contribute to the propagation of adipose inflammation[21]. Although B cells constituted the largest immune cell population within the BAT niche, their proportions were not affected by housing temperature.

Additionally, mice housed at colder ambient temperatures showed larger proportions of lymphatic ECs (Wald test $p = 0.0278$) (Supplementary Fig. 1p), non-myelinating Schwann cells (Wald test $p = 0.0041$) (Supplementary Fig. 1t), pericytes (Wald test $p = 0. 0466$) (Supplementary Fig. 1r), and lower proportions of VSMs (Wald test $p = 0.0305$) in BAT (Supplementary Fig. 1q). This is an intriguing observation, especially considering the recent identification of Trpv1-expressing cells derived from the VSM as the source of cold-induced thermogenic

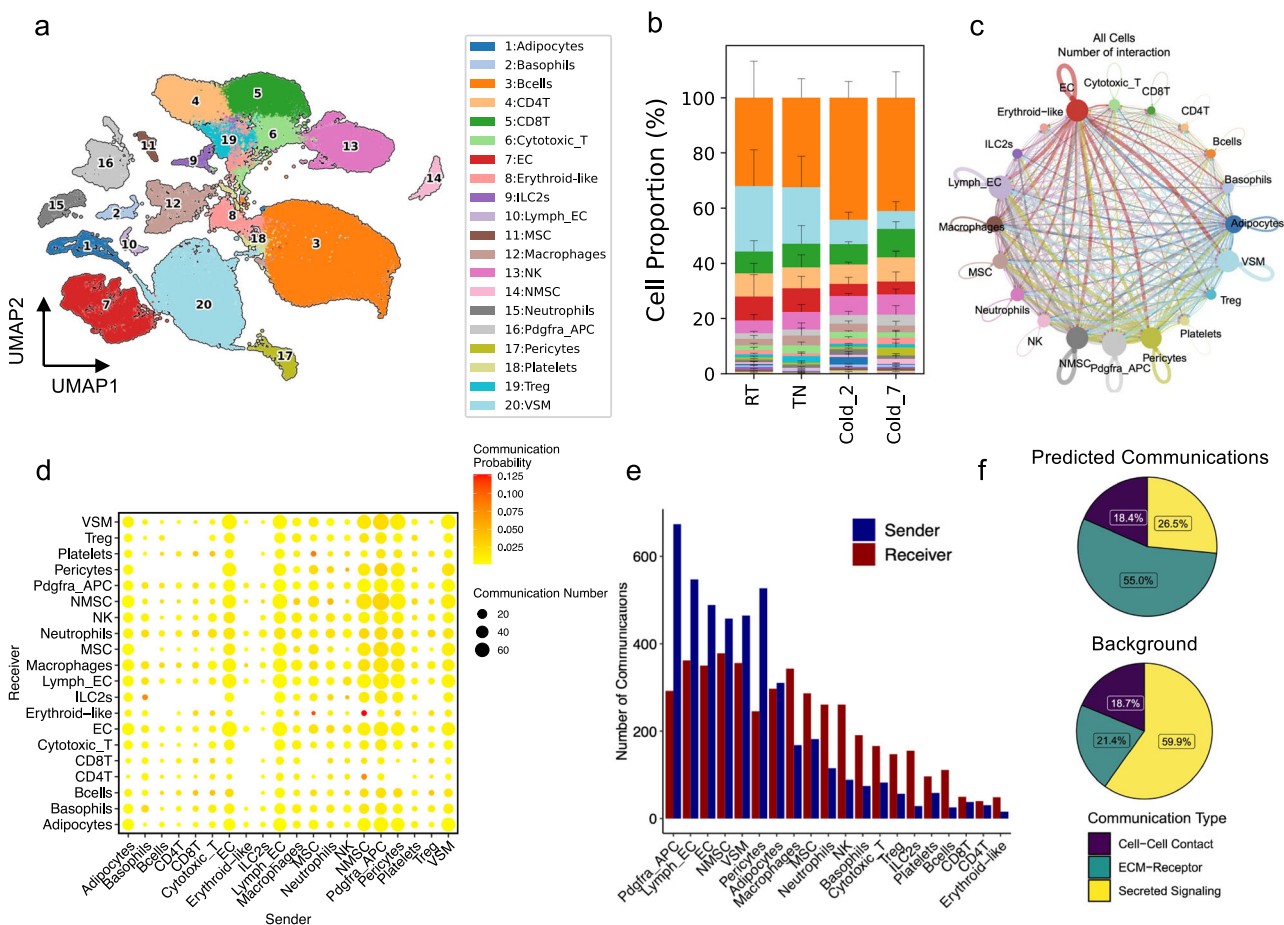

**Fig. 1 Overview of cellular composition and cell–cell interactions in BAT. a** Unsupervised clustering of cells from the BAT-SVF of 9-week-old male C57BL/6 J mice housed at TN (30 °C, 7 days), RT (22 °C), Cold (5 °C, 2 and 7 days) represented on a UMAP (N = 4/group). **b** The percentage of cells in each cell type under different housing conditions. The average percentage of replicate samples and their standard deviation are shown. **c** Circle plot showing the cell-to-cell communications in BAT. The line width represents the number of ligand-receptor interactions between two cell types. The size of the circle represents the number of interactions in each cell type, line width represents the number of interactions between two cell types, and color distinguishes sender cell types. **d** Dot plot showing the number of significant interactions between each pair of cell types. The size and color of the circle both represent the number and probability of interactions, respectively. **e** The number of significant communications involving each cell type as "sender" or "receiver". **f** The proportion of communications in cell–cell contact, ECM-Receptor, and Secreted Signaling categories among the predicted communications (top) and background (bottom).

adipocytes[9]. The changes in the proportions of various vascular cell populations also reflect the dynamic remodeling of BAT vasculatures in response to environmental temperature changes.

**Quantitative inference of the intercellular communications in BAT microenvironment.** Intercellular communication enables the individual cells in tissues to coordinate their functions and orchestrate development, homeostasis, and remodeling. Beyond cataloging the cellular composition of tissues, the cell type-specific transcriptional signature can be used to probe the intercellular interactions mediated by secreted and membrane-bound proteins[22]. To map the intercellular crosstalk in BAT mediated by protein-ligand and receptor interactions, we used the CellChat algorithm[23]. CellChat uses a database of more than 2000 interactions among ligands, receptors, and their cofactors representing the known heteromeric molecular complexes to infer the potential communications between cell types in scRNA-seq data. CellChat takes cell-type-specific gene expression data as input and computes the likelihood of cell–cell interaction by integrating gene expression with prior knowledge of the interactions between known ligands and receptor complexes. The main advantage of CellChat over other similar methods is that it considers the

composition of ligand-receptor complexes, including multimeric ligand and receptor complexes and cofactors such as soluble agonists, antagonists, co-stimulatory and co-inhibitory membrane-bound receptors.

CellChat identified a total of 4438 significant ligand-receptor pairs among the 20 cell groups. We used these interactions to construct the network of intercellular communications among all cell types (Fig. 1c, d). Constructing a weighted directed graph of significant connections between the cell types indicated that Pdgfra+ adipocyte progenitor cells (APC) were involved in the most significant number of interactions with other cell types (Fig. 1c, e). Pdgfra+ APC sent and received signals via 674 and 292 ligand-receptor pairs, respectively. These findings indicate adipocyte progenitors may serve as the dominant communication "hub" in the adipose niche and suggest the multifaceted roles of adipocyte progenitors in the adipose microenvironment beyond supporting adipogenesis. Vascular cells, including lymphatic and vascular endothelial cells, VSMs, and pericytes, are also engaged in a large number of communications with other cell types in the adipose niche (Fig. 1c, e).

Classifying the intercellular communications revealed that most crosstalks in BAT-SVF were mediated by the interactions of

cells with extracellular matrix (ECM) proteins (55% of the identified communications) while signaling through the secreted molecules and cell–cell contact made up 26.5% and 18.4% of the communications, respectively (Fig. 1f). This distribution was different from the background dataset in which secreted signaling, ECM-receptor interactions, and cell–cell contact represents 59.9%, 21.4%, and 18.7% of interactions, respectively (Fig. 1f). This is in agreement with the crucial roles of ECM components in the regulation of adipose tissue function and remodeling[24, 25] and suggests that the interaction of cells with the surrounding ECM in BAT is an important regulator of their functions.

To further verify these findings, we applied CellChat to another single-cell transcriptomics dataset of mouse BAT-SVF[26]. Burl and colleagues performed scRNA-seq of iBAT stromal cells in mice housed at room temperature (RT; 22–23 °C) or exposed to 6 °C for 4 days. BAT stromal cells were divided into immune and non-immune cell populations by magnetic bead cell separation (MACS) using a lineage marker cocktail containing anti-CD5, anti-CD11b, anti-CD45R (B220), anti-Gr-1 (Ly-6G/C), anti-Neutrophil(7/4), and anti-Ter-119 antibodies (Supplementary Fig. 2a, b). To map the intercellular crosstalk in this dataset, the lineage positive and lineage negative libraries were combined and used as input to the CellChat algorithm. Quantifying the total number of significant connections between the cell types indicated that the adipocyte stem and progenitor cells (ASC1-3: adipose tissue stromal cell, Prolif/Diff: proliferating/differentiating cells) were involved in the highest number of interactions with other cell types (Supplementary Fig. 2c). This is consistent with the notion that adipocyte progenitors serve as the primary communication center, or "hub", in the adipose tissue environment.

**Cold exposure promotes intercellular communications in BAT.** To determine how ambient temperature affects intercellular communications in BAT, we applied CellChat to analyze cell–cell interactions in each group (temperature) separately. Decreasing the environmental temperatures increased the total number of interactions in the BAT microenvironment (3586 in TN, 3246 in RT, 4251 in cold2, and 5281 in cold7, $p$ value < 2.2e-16) (Fig. 2a, b). This was reflected in significant increases in the number of both incoming and outgoing interactions from and to several cell populations. For example, 7 days of cold exposure increased the number of incoming and outgoing interactions to and from the Pdgfra+ APC, lymphatic and vascular endothelial cells, non-myelinated Schwann cells (NMSC), and myelinated Schwann cells (MSC), adipocytes, and several immune cell types (Macrophages, neutrophils, natural killer cell, ILC2s, etc.). Interestingly, the analysis also indicated that the Pdgfra+ APC received the highest number of incoming signals after 7 days of cold. At that time point, NMSC, VSM, and EC were the cell types that sent out significant outgoing signals.

Cold exposure stimulates a coordinated and multilayered remodeling of BAT, including the recruitment of brown adipocytes, expansion of vascular endothelial cells, sympathetic nerve outgrowth, and changes in the makeup of BAT-resident immune cells[6,9, 27]. The profound increase in the number of interactions among different cell types in the BAT indicates the higher need for effective communications between cells to coordinate their functions in response to thermogenic stimuli.

**Temperature-regulated remodeling of intercellular interactions in BAT.** Next, we analyzed the effect of housing temperature on the expression of individual ligands and receptors mediating crosstalk in BAT. Using K-means clustering on

communication probabilities of 125 most variable communication events across conditions, we identified five distinct patterns for temperature-dependent regulation of interactions (Supplementary Fig. 3a). Among those, interactions that follow Pattern 1 (upregulated in 7 days cold vs. other groups), Pattern 2 (upregulated with decreasing temperature across all groups), and Pattern 4 (downregulated with decreasing temperature across all groups) were most likely to be involved in adaptive BAT thermogenesis. The 46, 31, and 24 interactions representing patterns # 1, 2, and 4 are presented in Supplementary Fig. 3b, respectively.

Analyzing the expression of individual ligands revealed that cold exposure regulated the RNA expression level of cytokines or chemokines (*Ccl6, Ccl21a, Ccl5, Mif, Cxcl12, Cxcl2, Cxcl1*), ECM proteins (*Col4a1, Col1a1, Col1a2, Col4a2, Col6a1, Col6a2, Col6a3, Hspg2, Fn1, Tnxb, Ptn*), and cell adhesion proteins (*Selplg, L1cam, Ncam1, Pecam1, Esam1, Mpz*) in immune cells, vascular cells, and Schwann cells (Fig. 2c). Furthermore, cold exposure induced changes in the expression of multiple receptors on adipose-resident cells (Fig. 2d). For example, cold exposure significantly increased the expression of *Cd44* and *Cxcr2* in neutrophils, while it reduced the expression of Cd74 in macrophages (Fig. 2d). Other significantly regulated ligands and receptors in the vascular cells included the upregulation of *Esam* and *Col4a1/2* in pericytes, *Itga1* and *Itgb1* in VSMs, and *Pecam1* and *Fn1* in vascular endothelial cells (Fig. 2c, d). Collectively, these changes could shift how these niche components interact with each other to enable the functional and structural adaptation required for BAT thermogenesis.

**Crosstalk between adipogenic and immune cells.** Adipose tissue depots house a wide array of immune cells, including macrophages, B and T lymphocytes, NK cells, neutrophils, eosinophils, basophils, and ILC2s. Emerging evidence revealed several roles for the resident immune cells in adipose thermogenesis. Our scRNA-seq data identified several distinct populations of immune cells in BAT and enabled us to analyze the effect of housing temperature on the abundance and transcriptome of each population. To understand the ligand-receptor crosstalk between the immune cells and adipogenic cells, we mapped the interactions between the individual immune cell populations present in the scRNA-seq data (Basophils, B cells, CD4 T cells, CD8-T cells, Cytotoxic T cells, ILC2s, Macrophages, Neutrophils, NK cells, and Tregs) and adipogenic cells (adipocytes, Pdgfra+ APC, and VSM) (Fig. 3a). Notably, the Pdgfra+ APCs acted as major sender cells to communicate with macrophages, NK cells, and neutrophils. Among all the interactions between adipogenic and immune cells, 43% were between ECM and receptors, 32.9% were through secreted factors, and 24.1% were mediated via cell–cell contact (Fig. 3b).

This analysis also identified many interactions between the Pdgfra+ adipocyte progenitors and several immune cells. For example, Pdgfra+ adipocyte progenitors secrete amyloid beta (A4) precursor protein (App) that can bind to Cd74 on macrophages and B cells (Fig. 3c). Stimulation of CD74 can trigger the activation of several signal transduction cascades that play important roles in cell proliferation and survival[28]. Additionally, Pdgfra+ adipocyte progenitors secrete multiple types of collagens, such as Col1a1, Col1a2, Col4a1, Col4a1, Col6a1, Col6a2, and Col6a3, all of which could interact with Cd44 on macrophages or neutrophils (Fig. 3c). Cd44 is a transmembrane glycoprotein expressed in a variety of cell types. Cd44 interacts with extracellular matrix components and regulates the migration and recruitment of leukocytes[29].

Differential expression analysis across the four conditions (TN, RT, cold 2 days, and cold 7 days) identified several ligands and

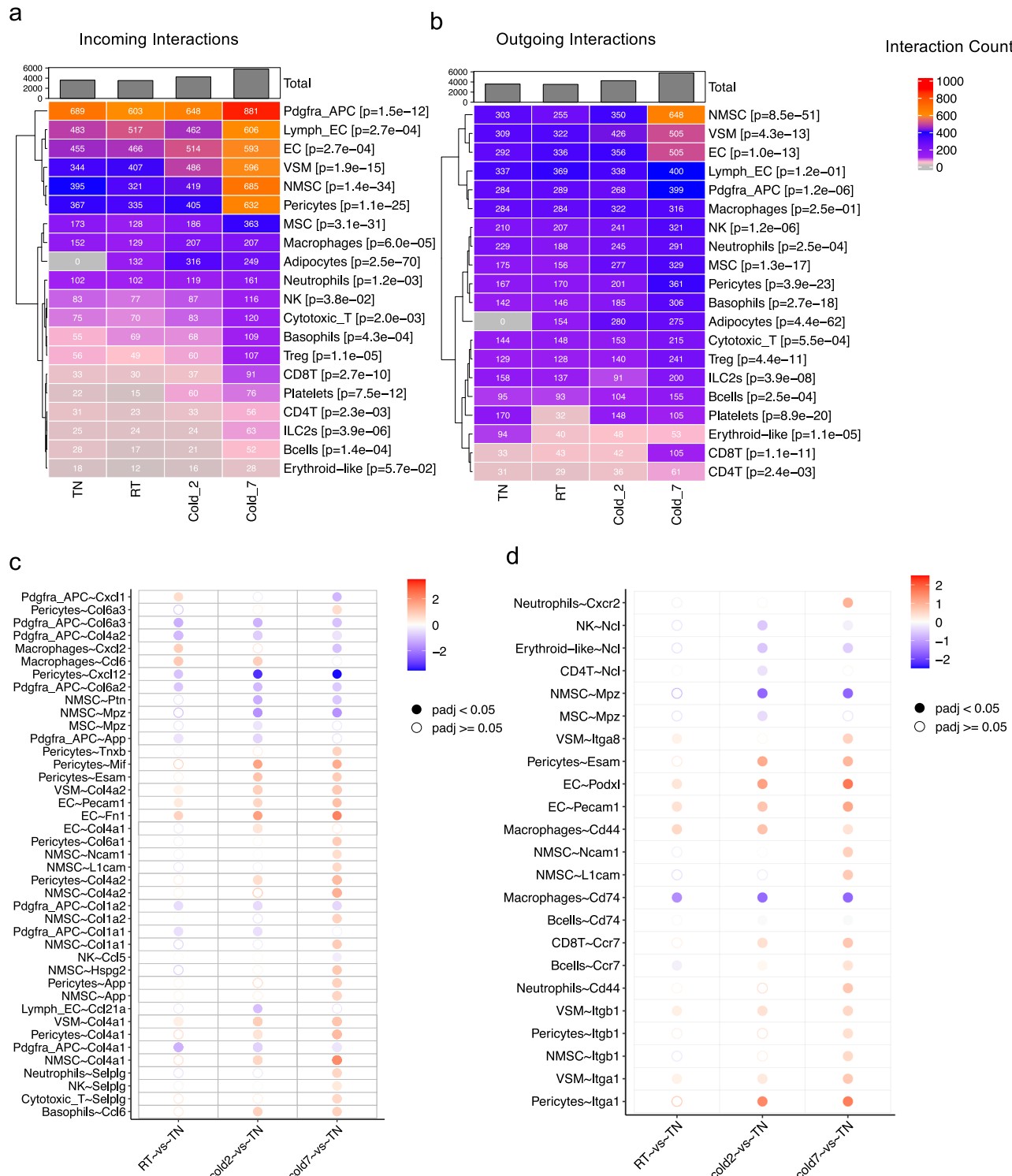

**Fig. 2 Temperature-regulated remodeling of intercellular interactions in BAT.** Heatmaps show the total number of significant (**a**) incoming and (**b**) outgoing interactions for each cell type across different conditions. The expression folds change of (**c**) ligands and (**d**) receptors relative to the TN (30 °C, 7 days) group. The color indicates the log2 fold change of gene expression between two conditions. The solid dots indicate significant differential expression (*p* value < 0.05) and the circle dots indicate non-significance.

receptors whose expression was modulated by cold, resulting in a significant decrease or increase in the interactions as defined by the calculated communication scores. For example, cold exposure significantly reduced the expression of transcripts encoding the ECM components (*Col6a2, Col6a1, Col6a3, Col1a1,* and *Col1a2*) in

Pdgfra+ adipocyte progenitors and its interacting partner, syndecan 4 (*Sdc4*) in macrophages (Fig. 3d). Furthermore, the reduced expression of C-C motif chemokine ligand 2 (*Ccl2*) in Pdgfra+ adipocyte progenitors could attenuate the proinflammatory Ccl2-Ccr2 signaling axis in Ccr2 expressing ILC2s in cold (Fig. 3d).

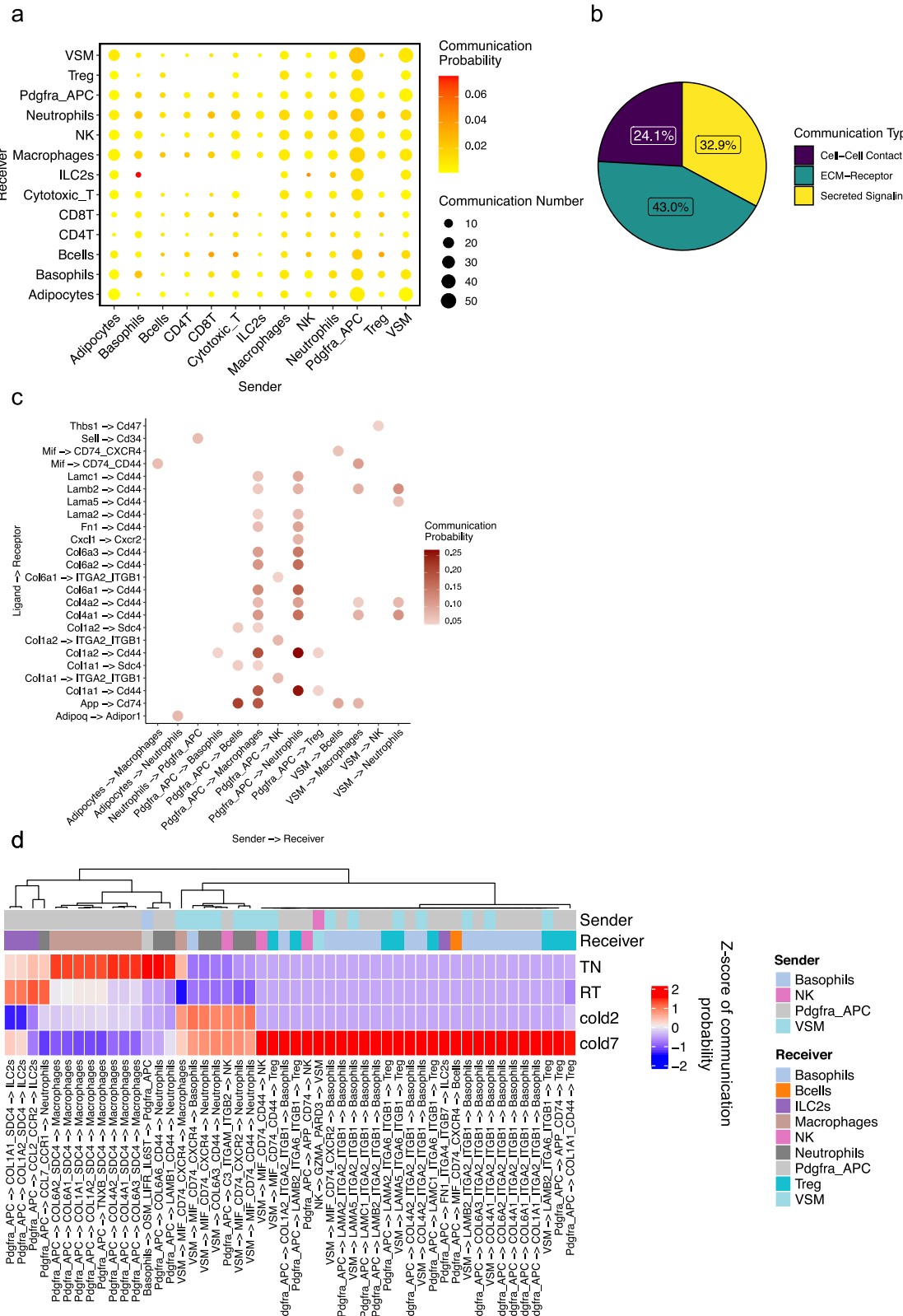

**Fig. 3 Crosstalk between adipogenic and immune cells. a** Dot plot showing the number of significant interactions between each pair of cell types in BAT-SVF of mice housed at TN, RT, and cold (2 and 7 days). The size and color of the circle both represent the number and probability of interactions, respectively. **b** The proportion of communications in Cell–Cell contact, ECM-Receptor, and Secreted Signaling categories among the predicted communications. **c** Dot plot presenting the most significant ligand-receptor interaction pairs between adipogenic and immune cells in BAT. The color of the circle represents communication probability. **d** Heat map showing the communication scores for each ligand–receptor interaction in the cell type pairs across different conditions.

Conversely, the interactions between the Macrophage migration inhibitory factor (MIF) ligand expressed by the VSMs and the receptor complexes Cd74-Cxcr4, Cd74-Cxcr2, and Cd74-Cd44 expressed on macrophages, basophils, neutrophils, NK, and Treg cells were predicted to increase in BAT of mice housed in cold (Fig. 3d). Mif is a pleiotropic cytokine that regulates leukocyte recruitment and plays a role in innate and adaptive immunity[30]. Another example is the increased interactions of the Pdgfra+ adipocyte progenitors with basophils and Tregs via laminins (Lama2, Lamc1, and Lamb2) and integrins (Itga2, Itgb1) (Fig. 3d).

**Crosstalk between adipogenic and vascular cells**. BAT is one of the most vascularized tissues in the body[31]. Environmental challenges, such as ambient cold, excess calorie intake, and physical activity, promote the expansion of adipose tissue vasculature primarily through inducing sprouting angiogenesis[32]. The expansion and remodeling of the vascular network are essential for providing the optimal supply of oxygen, nutrients, and other bioactive molecules for adipocytes. To understand the reciprocal communications between the cells that form adipose vasculature and adipogenic cells, we next focused on the interactions between vascular cells (vascular endothelial cells, lymphatic endothelial cells, VSM, and pericytes) and adipogenic cells (adipocytes, Pdgfra+ adipocyte progenitors, and VSMs) (Fig. 4a). Interestingly, more than half (62%) of the identified interactions were between ECM and their receptors, 25% were through secreted factors, and 13% occurred via cell–cell contact (Fig. 4b).

Adipocytes secrete angiogenic factors, including members of the vascular endothelial growth factor (VEGF) family that regulate angiogenesis through two tyrosine kinase receptors, VEGFR1 (encoded by *Flt1*) and VEGFR2 (encoded by *Kdr*)[32]. CellChat identified significant and specific crosstalk between adipocytes and endothelial cells mediated by the interaction of Vegfa and Vegfb with Flt1, Kdr, and the Flt1-Kdr receptor complex (Fig. 4c). Pdgfra+ adipocyte progenitors engaged in multiple crosstalks with vascular and lymphatic endothelial cells and pericytes. Most of these interactions appeared to occur through collagens and integrins (such as α1β1 and α9β1) expressed in the vasculature (Fig. 4c).

As expected, cold exposure robustly enhanced the interactions between adipogenic and vascular cells (Fig. 4d). Cold exposure significantly increased the expression of *Itga9* and *Itgb1* in vascular endothelial cells, potentially resulting in a higher abundance of Integrin α9/β1 complex in cold. This was predicted to enhance their interactions with Pdgfra+ adipocyte progenitors and VSMs through the ECM components such as Col6a3, Lamc1, Col1a1, Col6a1, Lamb2, Col4a1, Col4a2, Lama4, and Col6a2.

Cold exposure also induced the expression of Notch ligands, *Jag1* and *Jag2*, in endothelial cells and *Notch3* in VSMs, thus increasing the likelihood of Jag1/Jag2-Notch3 signaling between the vascular endothelial cells and VSMs in cold. The Jag-Notch signaling controls vascular development in embryonic and adult tissues. Specifically, Endothelial cell-derived Jag1 is essential for blood vessel formation by enhancing the differentiation and maturation of VSM cells[33, 34].

**Crosstalk between adipogenic and Schwann cells**. BAT is innervated by an extensive network of sympathetic and sensory nerve projections that are responsible for transmitting information between BAT and the central nervous system (CNS)[35]. Cold exposure increases sympathetic activity in BAT by elevating the rate of norepinephrine turnover and increasing the density of sympathetic arborizations[5, 36]. The communication between thermogenic adipocytes and sympathetic nerve plays an essential role in establishing the sympathetic network during the early postnatal period[37].

Although the innervating sympathetic nerves are not captured in the single-cell transcriptome analysis, we identified two distinct populations of myelinating and non-myelinating Schwann cells (NMSC and MSC) in BAT (Fig. 1a). Schwann cells are the major glial cells of the peripheral nervous system and are essential for the development, function, and regeneration of peripheral nerves[38]. Consistent with their roles in supporting axon growth, we found a significant increase in the frequency of NMSCs in BAT of mice housed in cold (Supplementary Fig. 1n). Analysis of ligand-receptor interactions identified many interactions between adipogenic cells (adipocytes, Pdgfra+ adipocyte progenitors, and VSMs) and Schwann cells (NMSC and MSC) (Fig. 5a). Most of these communications were mediated by ECM-receptor interactions (73%). Signaling by secreted ligand and direct cell–cell contact made up 21% and 6% of total interactions, respectively (Fig. 5b).

The largest interactions in this category were between Pdgfra+ adipocyte progenitors and NMSCs. Examples of such interactions included the binding of several ECM proteins expressed in Pdgfra+ adipocyte progenitors (Col1a1, Col1a2, Col4a1, Col6a1, Col6a2, Col6a3, and Tnxb) to syndecan 4 (Sdc4) expressed by MSCs and NMSCs (Fig. 5c). Additionally, adipocytes and APCs expressed Prosaposin (Psap) that binds to Gpr37l1 present in NMSCs (Fig. 5c). Psap exhibits neurotrophic and myelinotrophic activities in neurons and glial cells[39,40].

Among the interactions that were differentially regulated by temperature, most were through integrin receptor complexes: α2β1 and α5β8 on NMSCs, α5β8 on MSCs, α1β1 and α3β1 on VSMs (Fig. 5d). Cold exposure increased the expression of *Itga1*, *Itga3*, and *Itgb1* in VSMs, likely facilitating their interaction with Pdgfra+ adipocyte progenitors through Col4a1, Col6a1, Col6a2, Col1a2, Col4a1, Col1a1, and Col4a2 (Fig. 5d).

# Discussion

Applying the CellChat method to single-cell transcriptomic data of mouse BAT-SVF, we present a comprehensive map of ligand-receptor interactions in the thermogenic adipose tissue across four different housing temperatures and time points. Our analysis reveals that BAT adaptation to environmental temperature involves major changes in the cellular composition as well as the rewiring of the intercellular crosstalk. The quantitative comparison of the ligand-receptor interactions across different temperatures showed that cold exposure promotes intercellular communications in BAT, suggesting the increased need for information exchange among cells to coordinate the tissue adaptation to the rise in energetic demand. This is in line with a recent study showing that obesity and HFD increased the intensity of ligand-receptor pair expression in the white adipose tissue of humans and mouse[41, 42]. These findings highlight the role of intercellular communications in adipose tissue remodeling in response to fluctuating nutritional status and metabolic demand.

Despite great advancement in the characterization of adipocyte progenitors and their contribution to the expansion and turnover of the adipocyte pool, their roles beyond adipogenesis remain poorly understood. Our systematic analysis of ligand-receptor interactions identified adipocyte progenitors as the dominant communication hub in the thermogenic adipose niche. Comparing the transcriptome of adipocyte progenitors across different temperatures demonstrates that housing temperature regulates the transcription of ligands and receptors involved in ECM remodeling, immune modulation, angiogenesis, and neurogenesis. These findings underscore the diverse regulatory functions of adipocyte progenitors in adipose tissue beyond adipogenesis.

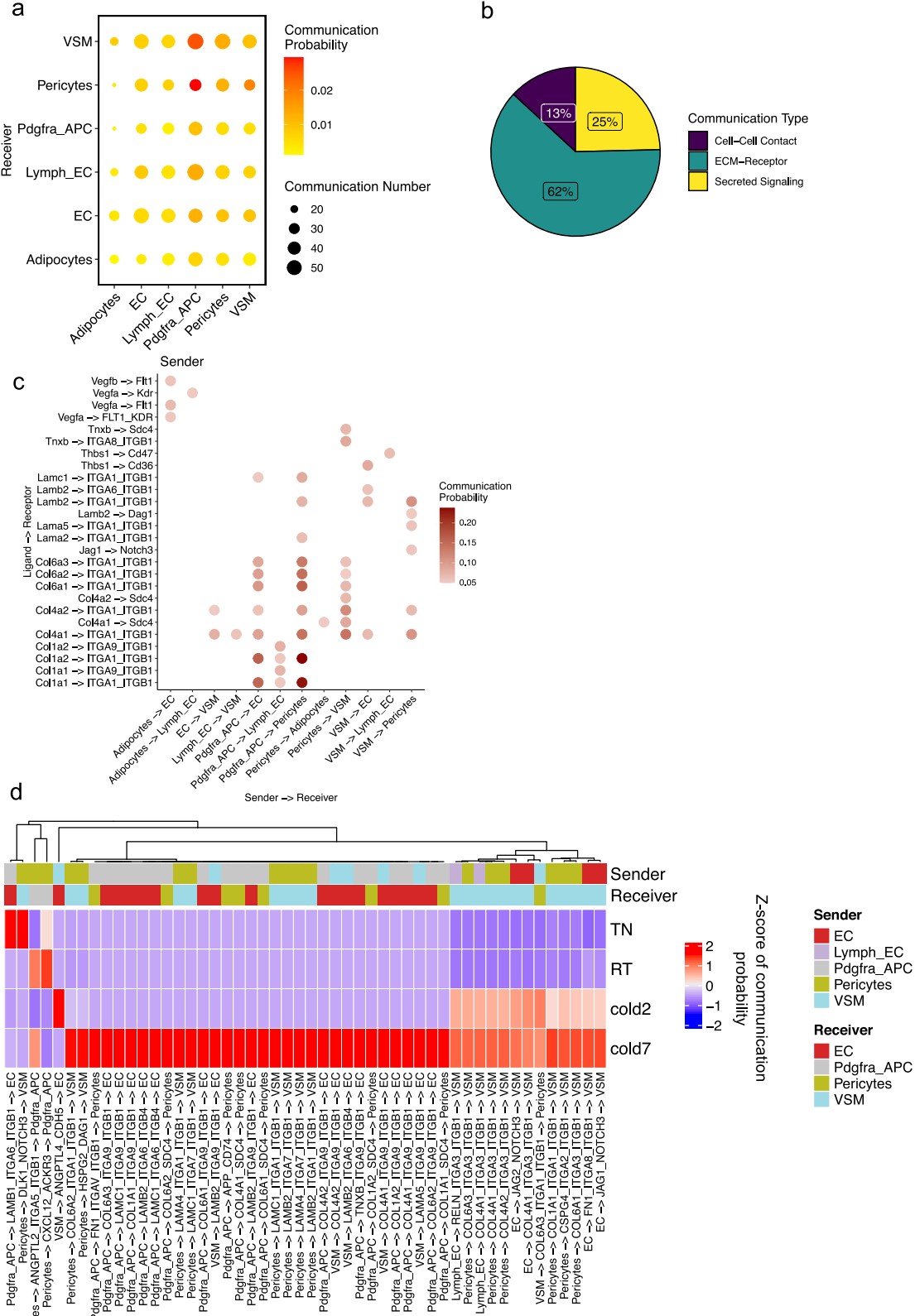

**Fig. 4 Crosstalk between adipogenic and vascular cells. a** Dot plot showing the number of significant interactions between each pair of cell types in BAT-SVF of mice housed at TN, RT, and cold (2 and 7 days). The size and color of the circle both represent the number and probability of interactions, respectively. **b** The proportion of communications in cell–cell contact, ECM-Receptor, and Secreted Signaling categories among the predicted communications. **c** Dot plot presenting the most significant ligand-receptor interaction pairs between adipogenic and vascular cells in BAT. The color of the circle represents communication probability. **d** Heat map showing the communication scores for the top 50 ligand-receptor interaction in the cell type pairs across different conditions.

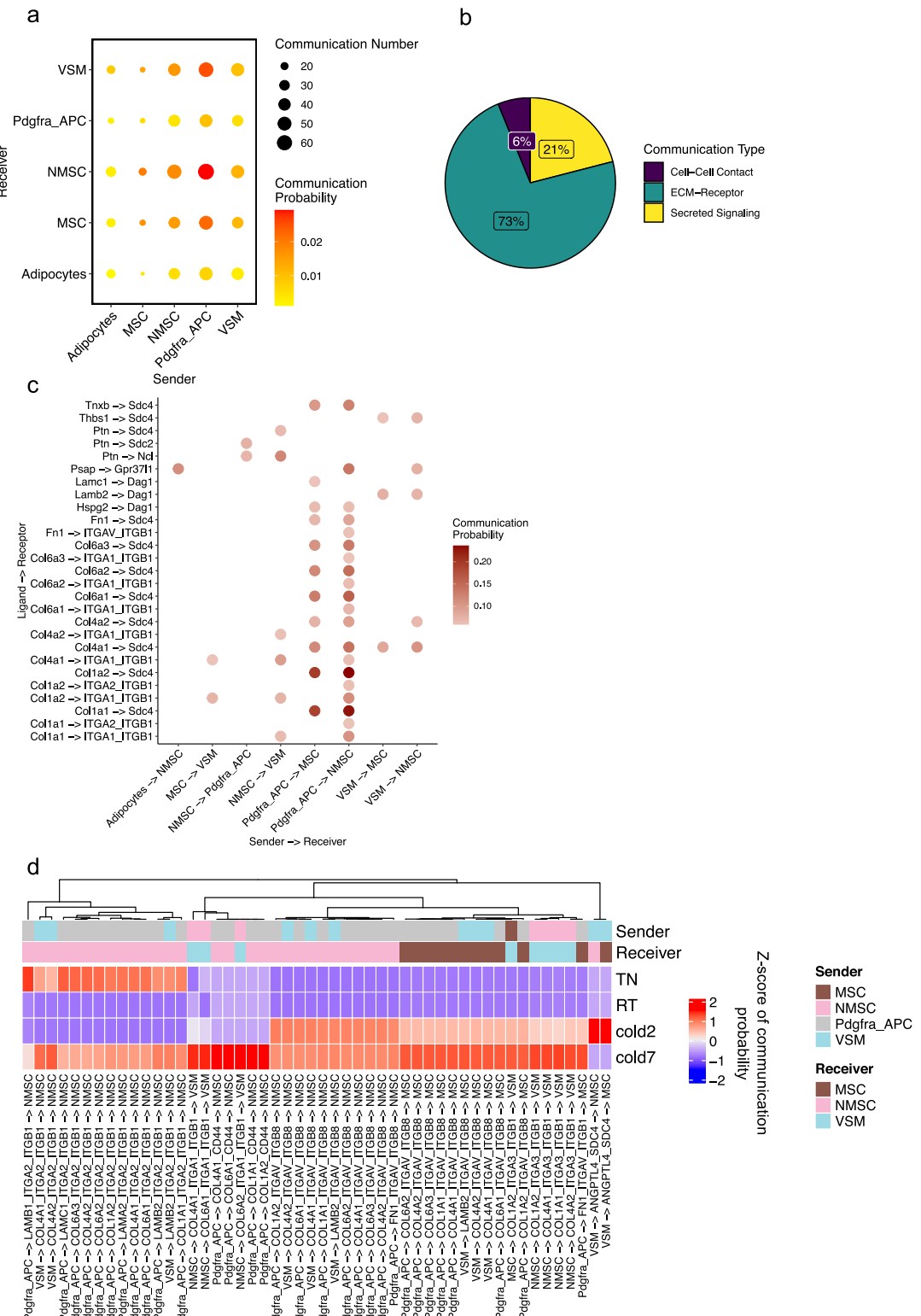

**Fig. 5 Crosstalk between adipogenic and Schwann cells. a** Dot plot showing the number of significant interactions between each pair of cell types in BAT-SVF of mice housed at TN, RT, and cold (2 and 7 days). The size and color of the circle both represent the number and probability of interactions, respectively. **b** The proportion of communications in cell–cell contact, ECM-Receptor, and Secreted Signaling categories among the predicted communications. **c** Dot plot presenting the most significant ligand-receptor interaction pairs between adipogenic and Schwann cells in BAT. The color of the circle represents communication probability. **d** Heat map showing the communication scores for the top 50 ligand-receptor interaction in the cell type pairs across different conditions.

Collectively, our integrative analysis of intercellular crosstalk in BAT offers a comprehensive understanding of the mechanisms that control thermogenesis. The database of ligand-receptor interactions identified in this study offers a valuable platform for future investigations of the functions of specific signaling molecules and cell types in BAT development and function.

## Limitations of the study

One limitation of the study is that intercellular interactions in the BAT-SVF exclude most adipocytes. Obtaining intact single adipocytes for transcriptome profiling along the entire adipogenic trajectory at the single-cell level is a major challenge due to the large size, fragile nature, and high buoyancy of mature adipocytes. Alternative methods have been used to overcome this challenge, such as isolating nuclei from fresh or frozen adipose tissue. However, nuclear transcripts only represent a fraction of the cellular transcriptome. Additionally, comparing single nuclei with whole-cell transcriptomes of human adipocytes and pre-adipocytes has revealed inherent transcript enrichment and detection biases in the single-nucleus RNA sequencing (snRNA-seq)[43]. Therefore, single-cell and single-nucleus transcriptomic methods are complementary approaches with specific applications in studying adipose tissue.

Another limitation of single-cell transcriptome studies is the possibility of transcriptional changes induced during the tissue processing and digestion steps. The difference in the fragility and size of different cell types might result in the loss or mis-representation of some cell types. Moreover, these changes can also introduce technical artifacts in the transcriptome data, making it difficult to distinguish actual biological variation from technical variation, especially among different studies generated in different labs. In this study, we have included four biological replicates in each group and carefully considered the tissue processing methods to ensure accurate and meaningful interpretation of single-cell transcriptome data.

While our integrative analysis of intercellular crosstalk in BAT provides a comprehensive understanding of the mechanisms involved in thermogenesis, it is important to note that the ligands and receptors identified in this study are based on computational prediction. Future experimental approaches using genetic and pharmacological manipulations are necessary to test the functional relevance of these findings.

## Methods

**Animals**. All animal experiments were performed in compliance with all relevant ethical regulations for the testing and use of small rodents and with approval by the Institutional Animal Care and Use Committees (IACUC) at Joslin Diabetes Center. 9-week-old male C57BL/6 J mice (Stock no. 000664) were purchased from The Jackson Laboratory and were used for the scRNA-sequencing experiment.

Mice were housed at room temperature (22 °C), cold (5 °C for 2 days and 7 days), or thermoneutral (30 °C for 7 days) in controlled environmental diurnal chambers (Caron Products & Services Inc., Marietta, OH) with free access to food and water. Mice were maintained at a 12 h-light/dark cycle with 30% Humidity on a normal chow diet containing 22% of calories from fat, 23% from protein, and 55% from carbohydrates (Mouse Diet 9 F 5020; PharmaServ). The procedures for the isolation of the stromal vascular fraction from BAT and single-cell RNA-sequencing were previously described[9] and outlined below.

**Isolation of the stromal vascular fraction from BAT**. Interscapular BATs from four animals per group were dissected, minced, and digested with a cocktail containing type 1 Collagenase (1.5 mg/ml; Worthington Biochemical), Dispase II (2.5 U/mL; STEMCELL Technologies), fatty acid-free bovine serum albumin (2%; Gemini Bio Products) in Hanks' balanced salt's solution (Corning® Hank's Balanced Salt Solution, 1× with calcium and magnesium) for 45 minutes at 37 °C with gentle shaking. Dissociated tissue was centrifuged at $500 \times g$ at 4 °C for 10 minutes. The top adipocyte layer and the supernatant were gently removed, the SVF pellet was resuspended in 10 ml of 10% Fetal Bovine Serum (FBS) in DMEM, filtered through a 100 μm cell strainer into a fresh 50 ml tube, and centrifuged at $500 \times g$ for 7 minutes. The red blood cell lysis was performed by resuspending the pellet in 2 ml sterile ACK (Ammonium-Chloride-Potassium) lysis buffer (ACK

Lysing Buffer, Lonza) and incubating on ice for 5 minutes. The cells were then filtered through a 40 μm cell strainer, washed with 20 ml 10% FBS in DMEM, and centrifuged at 500 g for 7 minutes. The pellet was resuspended in 1 ml of 1.5% BSA in PBS. The dead cell removal was performed using Dead Cell Removal Kit (Miltenyi Biotec) according to the manufacturer's instructions. The cells were finally resuspended in 50–100 μl of 1.5% BSA in PBS and kept on ice before immediately proceeding to single-cell isolation. The detailed protocol for the isolation of the SVF from adipose tissue is deposited on protocols.io (https://doi.org/10.17504/protocols.io.bpurmnv6).

**Single-cell RNA-sequencing**. Cells were loaded for an expected recovery of 10,000 cells per channel. The chip loaded with single-cell suspension was placed on a 10x Genomics Chromium Controller Instrument (10× Genomics, Pleasanton, CA, USA) to generate single-cell droplets containing uniquely barcoded GEMs (Gel Bead-In Emulsions). Single-cell RNA-seq libraries were obtained following the 10x Genomics recommended protocol, using the reagents included in the Chromium Single-Cell 3′ v3 Reagent Kit. The libraries were sequenced on the NovaSeq S2 flow cell (Illumina, 100 cycles).

**Versions of software and packages**. Cellranger 6.1.2, Python 3.8; Python packages include scanpy 1.8.2, pandas 1.4.1, matplotlib 3.5.1, seaborn 0.11.2, jupyter 1.0.0; R 4.1.1, R packages include CellChat 1.1.3, ggplot 3.3.5, ggrepel_0.9.1, ComplexHeatmap_2.8.0.

**Single-cell RNA-seq data analysis**. The Cellranger was used to align single-cell RNA sequences to the mouse reference genome (mm10) and obtain the read count for each gene in each cell. The outputs in read count in each gene and each cell from Cellranger were provided to Scanpy for further processing. This processing includes filtering low-quality cells and genes, expression normalization, dimensional reduction, clustering, and visualization of gene expression. Cells were filtered if they had <800 or >50,000 UMIs and if they had <400 or >7500 detected genes. Genes were filtered if they were detected in <10 cells. Total read counts in each cell were normalized to 10 thousand, and the normalized value of each gene in each cell was log-transformed to obtain the final normalized gene expression. 10 nearest neighbors and 40 principal components were used in clustering and UMAP analysis. Clustering was done by the Leiden algorithm. The visualization of the clusters was performed by the UMAP method (Fig. 1a). The clusters were annotated to cell types by known marker genes of cell types and the top differentially expressed genes in the cluster. The known cell type marker genes were collected from PanglaoDB at https://panglaodb.se/markers.html. Differential expression analysis was also performed by Scanpy for each cell type between any two conditions from TN, RT, cold2, and cold7.

A published scRNA-seq dataset of iBAT-SVF generated by Burl and colleagues was also analyzed in this study. Their preprocessed data (expression matrix) in MTX file format was downloaded from the NCBI GEO database under accession number GSE207707. The GSM6310683, GSM6310684, GSM6310685, GSM6310686, GSM6310689, GSM6310690, GSM6310691, and GSM6310692 samples were included in the analysis. The data were analyzed following the scripts released by the authors (available at https://github.com/RBBurl1227/eLife-2022-ColdInducedBrownAdipocyteNeogenesis). Briefly, the preprocessed data were subjected to the Seurat package (version 3.1.5) in R for data normalization, batch effect removal, dimensional reduction, clustering, and cell annotation.

**Cell composition analysis**. To investigate the cell type composition in BAT from mice housed at four conditions (TN, RT, cold2, and cold7), we calculated the cell proportion score, namely, the percentages of cells in a cell type over the total cell number in each sample in the condition. The cell proportion of each cell type and each condition was visualized by a stacked bar plot with standard deviation across replicate mice in Fig. 1b. Further, the proportion of a cell type in each mouse and each condition was also visualized by a boxplot. The Wald test was used to calculate the $p$-value to suggest the significance of cell composition change across four conditions by each cell type (Supplementary Fig. 1).

**Cell–cell communication inference in BAT by CellChat**. The normalized expression matrix and cell type annotation generated from Scanpy or Seurat were passed to CellChat for ligand-receptor cell–cell communication analysis. To quantitatively infer intercellular communications in BAT, all cells from four conditions (TN, RT, cold2, and cold7) were pooled and grouped by cell types for CellChat analysis. Default parameters were used, except that min.cells were set to 10, which allows filtering out cell types with the total number of cells smaller than 10 in CellChat. Mouse ligand-receptor database without any selection was used in CellChat. A circle plot was used to visualize the number of inferred communications between cell types by the netVisual_circle function in the CellChat package (Fig. 1c). One communication was counted by a unique combination of sender cell type, ligand, receptor, and receiver cell type.

**Identification of temperature-regulated cell–cell communications**. Cells were grouped by cell types and conditions to perform CellChat analysis. To identify highly changed communications between conditions (TN, RT, cold2, and cold7), a communication-by-condition matrix was generated and filled by communication probability scores. Any communications with a $p$ value <0.05 in one of the four conditions were included in this matrix. Next, the mean and variance of communication probability were calculated for each row of the matrix, namely, each communication. The temperature-regulated communications were defined if their mean communication probability was greater than 0.05, and the variance greater than 0.0005. The two cutoffs for mean and variance probability were determined based on their distribution across all the communications, respectively. 126 ligand-receptor communications with the largest variation of communication probability across four conditions were selected and considered as temperature-regulated communications.

To better interpret those temperature-regulated communications, k-means clustering was used to group the identified 126 ligand-receptor communications. Five clusters were determined to represent five different patterns of communication across four conditions. To visualize the five patterns, communication probabilities of all ligand-receptor interactions in the pattern were averaged by each condition, which resulted in four average communications probabilities corresponding to four conditions in each pattern. Then, the averaged communication probabilities for each pattern were shown in a line graph (Supplementary Fig. 3a). Further, the communication probability of each communication in patterns was also shown in a heatmap (Supplementary Fig. 3b). In the analysis of temperature-regulated communications, adipocyte was not included, since it was mostly differentiating adipocytes, and few cells were included in the TN, cold2, and cold7 conditions.

**Statistics and reproducibility**. The Wald test was used to calculate the $p$ value of cell composition changes across different housing conditions. Other statistics were computed using the parameters provided by CellChat. Statistical methods were not used to predetermine sample sizes. Each independently isolated BAT-SVF sample from an individual mouse is considered a separate replicate.

**Reporting summary**. Further information on research design is available in the Nature Portfolio Reporting Summary linked to this article.

## Data availability

scRNA-seq data are deposited into the Gene Expression Omnibus database under accession number GSE160585 and are available at the following URL. Supplementary Data 1 includes the specifically expressed genes across 20 cell types identified from scRNA-seq dataset.

## Code availability

Scripts for data processing and downstream analyses are available through GitHub at https://github.com/zhengrongbin/mouseBAT_LR_paper.

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

## Acknowledgements

This project was supported in part by the National Institutes of Health (NIH) grants R01DK132469 and R01DK133528 (to Y.-H.T.), K01DK125608 and R03DK135786 (to F.S.), R01GM125632 and 1R01HL148338 (to K.C.), and P30DK036836 (to Joslin Diabetes Center's Diabetes Research Center) from the National Institute of Diabetes and Digestive and Kidney Diseases.

## Author contributions

F.S. and Y.-H.T. conceived the study, designed, and conducted the scRNA-seq experiment. R.Z. and K.C. conducted the scRNA-seq and CellChat analyses. L.-L.H. assisted in single-cell isolation and library preparation. F.S., Y.-H.T, R.Z., and K.C. wrote the manuscript. All authors read and approved the final manuscript.

## Competing interests

The authors declare no competing interests.
