## [Peer Review File · Communications Biology]

Comprehensive analysis of intercellular communication in the thermogenic adipose nicheReviewers' comments:

Reviewer #1 (Remarks to the Author):

In this manuscript, Shamsi et al. describe the results of applying the so-called CellChat algorithm to the single cell RNA-sequencing data (scRNA-seq) from brown adipose tissue (BAT) stromal vascular fractions of mice housed at three different temperatures, thermoneutral, room temperature, and cold. CellChat uses a database of thousands of known interactions among ligands, receptors, and cofactors to infer the potential interactions between cell types. The BAT SVF scRNA-seq data were previously used in Shamsi et al 2021 Nature Metabolism 3:485 for a clustering analysis that led to the identification of Trpv1 as a marker of thermogenic adipocyte progenitors. Hematopoietic cells were excluded in the 2021 clustering analysis but are included the present work, which now identifies 20 distinct clusters or cell types compared to 8 clusters in the 2021 work.

The use of the CellChat algorithm in the current work revealed interesting new information regarding the intercellular interactions that occur in BAT. For example, Shamsi et al have found that cold exposure enhances the intercellular interactions to and from the Pdgfra+ adipocyte progenitors, lymphatic and vascular endothelial cells, Schwann cells, adipocytes, and various immune cells such as macrophages, neutrophils, and type 2 innate lymphoid cells (ILC2). Numerous temperature-regulated ligand-receptor interactions based on the CellChat analysis were identified and analyzed in this study. Other interesting findings include that cells in BAT rely more extensively on receptor-extracellular matrix protein interactions than on secreted molecules and cell-cell contact, in contrast to the background dataset where secreted factors are most important.

Overall, this work is a valuable contribution that uses cutting edge technology to integrate large amounts of data and reveal important new insights about the diverse cellular components that make up BAT and the regulation of intercellular communication among these cell types by the environmental temperatures. Although the work consists almost entirely of bioinformatics analysis and includes little functional experimental data, it can form the basis for future studies of specific signaling molecules and cell types. In my opinion, this study is appropriate for publication in Communications Biology.

Reviewer #2 (Remarks to the Author):

The manuscript Shamsi et al. mines a previously published data set to extract possible cell-cell communication in brown adipose tissue of mice exposed to the cold for 2 or 7 days. The main concern is that the processes speculated to be mediated by the putative cell-cell communication not are demonstrated, nor is there any validation of the putative communication. As it stands, the analysis is rather speculative and the putative interactions too diverse to suggest follow up experimentation.

1) For readers that do not wish to evaluate the data independently, the authors should provide a list of the DEGs that were used to define cell types, including the % of cells expressing a particular transcript and fold-enrichment.

2) It is unclear how cell proportions were calculated (Fig 1B). Theoretically, due to differences in the

number of cells recovered in each library (~2,500-18,000), the libraries should all be clustered for cell identity and the proportions calculated. For example, # of individual cell type in library #1/total cells in library #1. It appears as if the cell proportions were calculated as number per library per total cells from all libraries in a given condition.

3) The authors should provide evidence supporting a role the processes proposed to be mediated by intercellular communication (angiogenesis, progenitor proliferation, sympathetic innervation), recognizing these have important temporal and spatial components.

4) The study design misses the major induction of BA neogenesis that occurs between 3 and 5 days of cold (Bukoweicki et al, Lee et al, 2015). This issue was raised in the recent publication (Burl et al., 2022) that also addressed cold-induced neogenesis by scRNA-seq. The Burl et al. (2022) and Shamsi data sets might be integrated to provide a more complete assessment

5) It is widely recognized that there are at least 2-3 ASC subtypes that express PDGFRA. The current data set does not distinguish among ASC subtypes, including adipogenic subtypes.

6) The authors suggest there is important cellular communication between mature adipocytes and a variety of other cell types. However, captured BA are speculated to be newly differentiated cells since they did not float. Inspection of the published scRNA libraries indicates that the BA phenotype is limited to only one of the 4 2-day-cold libraries, and none of the 7-day libraries (reflected in supplemental figure 1A), raising the issue of reproducibility. Supplemental figure 1A should show individual values of all libraries. At this point it would be difficult to make claims concerning secreted angiogenic factors from mature cells.

7) It is a little surprising that B and T lymphocytes are identified as contributing to over 50% of the stromal population. Nonetheless, it is barely mentioned what type of cell chat is involved in this population. As a major cell type, there should be more evaluation of how they may contribute to cold-induced neogenesis.

Reviewer #3 (Remarks to the Author):

In their manuscript Shamsi and colleagues apply a novel bioinformatic technique to single cell RNAseq data obtained from the BAT of mice kept at thermoneutrality and exposed to cold. Using the CellChat algorithm they find that changes in the ambient temperature elicit comprehensive alterations of cell-cell communications in murine BAT.

This is a very interesting manuscript and the findings are novel. However, several points should be revised and the conclusions drawn by the authors should be put into perspective.

Major points:

1. The authors analyzed the stromal-vascular fraction "SVF" of BAT samples from mice kept at different ambient temperatures for different amounts of time. In the manuscript the distinction between SVF and BAT is rarely made and conclusion are drawn for BAT as a whole. This is not true since mature adipocytes made up only a very small proportion of the analyzed cells. The reviewer is well aware of the fact that it is challenging to do scRNASeq of mature adipocytes but this is a major limitation of the paper and should be addressed appropriately.

2. I tried to figure out how many mice per treatment group were used but I couldn't find this data.
3. Could the treatment of the tissue (mincing, digesting, straining,...) prior to isolation of the RNA influence the expression levels of the analyzed genes which are involved in cell-cell interactions?
4. On which groups are the figures 3/4/5 a to c based: is this an analysis of all interventions groups (pooled) or only data from the animals kept at RT?
5. cross-talk between adipogenic cells and vascular cells
There does not seem to be much of a difference in terms of cell chatting between the thermoneutral and RT groups (Figure 4 d). Additionally, I would expect more similarities between the cold2 and cold7 groups.
6. This seems to be the case for the interactions between adipogenic cells and Schwann cells (figure 5 d).

Discussion / Conclusion

In general these data are interesting but they are largely descriptive in nature and do only rely on the transcriptome. As stated by the authors the software used makes predictions on cell-cell interactions and does not give unequivocal proof of the cell communications. The data can very well serve as a starting point to generate new hypotheses about BAT differentiation and regulation which need to be tested in the future. These limitations should be discussed appropriately.

line 371: "Collectively, our integrative analysis of intercellular crosstalk in BAT provides a holistic understanding of the mechanisms that control thermogenesis."

- This is a very bold statement given the rather descriptive nature of the findings in this paper. The following sentence is much closer to reality: it is "a valuable platform for future investigations"

Minor points:

Figure 2: the black dots should mark $p_{\text{adjusted}} < 0.05$ and not ≤ 0.05

Figure 3/4/5: Why did the authors use color and size to display the number of cell-cell interactions within the graphs A and C. This seems to be redundant.

We would like to extend our sincere appreciation to the editorial team and reviewers for dedicating time to reviewing our work and providing valuable and positive feedback. Their insights and comments have helped improve the manuscript. The manuscript has been revised to address the reviewers' comments and critiques as discussed in the point-by-point responses below.

Response to reviewers' comments:

Reviewer #1 (Remarks to the Author):

In this manuscript, Shamsi et al. describe the results of applying the so-called CellChat algorithm to the single cell RNA-sequencing data (scRNA-seq) from brown adipose tissue (BAT) stromal vascular fractions of mice housed at three different temperatures, thermoneutral, room temperature, and cold. CellChat uses a database of thousands of known interactions among ligands, receptors, and cofactors to infer the potential interactions between cell types. The BAT SVF scRNA-seq data were previously used in Shamsi et al 2021 Nature Metabolism 3:485 for a clustering analysis that led to the identification of Trpv1 as a marker of thermogenic adipocyte progenitors. Hematopoietic cells were excluded in the 2021 clustering analysis but are included the present work, which now identifies 20 distinct clusters or cell types compared to 8 clusters in the 2021 work.

The use of the CellChat algorithm in the current work revealed interesting new information regarding the intercellular interactions that occur in BAT. For example, Shamsi et al have found that cold exposure enhances the intercellular interactions to and from the Pdgfra⁺ adipocyte progenitors, lymphatic and vascular endothelial cells, Schwann cells, adipocytes, and various immune cells such as macrophages, neutrophils, and type 2 innate lymphoid cells (ILC2). Numerous temperature-regulated ligand-receptor interactions based on the CellChat analysis were identified and analyzed in this study. Other interesting findings include that cells in BAT rely more extensively on receptor-extracellular matrix protein interactions than on secreted molecules and cell-cell contact, in contrast to the background dataset where secreted factors are most important.

Overall, this work is a valuable contribution that uses cutting edge technology to integrate large amounts of data and reveal important new insights about the diverse cellular components that make up BAT and the regulation of intercellular communication among these cell types by the environmental temperatures. Although the work consists almost entirely of bioinformatics analysis and includes little functional experimental data, it can form the basis for future studies of specific signaling molecules and cell types. In my opinion, this study is appropriate for publication in Communications Biology.

We thank this reviewer for the time spent reviewing our manuscript and the positive feedback on our work.

Reviewer #2 (Remarks to the Author):

The manuscript Shamsi et al. mines a previously published data set to extract possible cell-cell communication in brown adipose tissue of mice exposed to the cold for 2 or 7 days. The main concern is that the processes speculated to be mediated by the putative cell-cell communication are not demonstrated, nor is there any validation of the putative communication. As it stands, the analysis is rather speculative and the putative interactions too diverse to suggest follow up experimentation.

We thank Reviewer #2 for the constructive critiques. While we agree with this reviewer that validation of the predicted communications is important, we want to clarify that the purpose of this manuscript is to provide a resource to the community rather than experimentally verifying the specific interactions. The established ligand-receptor interactome of BAT collected from mice housed at different temperatures provides an invaluable resource allowing other researchers to develop new hypotheses and design studies to test them.

1) For readers that do not wish to evaluate the data independently, the authors should provide a list of the DEGs that were used to define cell types, including the % of cells expressing a particular transcript and fold-enrichment.

We thank the reviewer for this suggestion. The list of differentially expressed transcripts in each cell type and the percentage of cells expressing those transcripts is now included in Supplementary Table 1.

2) It is unclear how cell proportions were calculated (Fig 1B). Theoretically, due to differences in the number of cells recovered in each library (~2,500-18,000), the libraries should all be clustered for cell identity and the proportions calculated. For example, # of individual cell type in library #1/total cells in library #1. It appears as if the cell proportions were calculated as number per library per total cells from all libraries in a given condition.

We agree with the reviewer that normalizing the cell number for each individual library would be more informative. The graph in Figure 1b is modified to show the cell type proportion relative to the total number of cells in each library.

3) The authors should provide evidence supporting a role for the processes proposed to be mediated by intercellular communication (angiogenesis, progenitor proliferation, sympathetic innervation), recognizing these have important temporal and spatial components.

In this revised manuscript, we have discussed the existing evidence in the literature that supports the role of identified ligands and receptors in the regulation of adipose tissue remodeling and thermogenesis. The revised manuscript includes a new section (lines 390-416 in the revised manuscript) in which we discuss this limitation and emphasize the need for future experimental approaches to address the function of the identified crosstalk in BAT physiology.

4) The study design misses the major induction of BA neogenesis that occurs between 3 and 5 days of cold (Bukoweicki et al, Lee et al, 2015). This issue was

raised in the recent publication (Burl et al., 2022) that also addressed cold-induced neogenesis by scRNA-seq. The Burl et al. (2022) and Shamsi data sets might be integrated to provide a more complete assessment

We thank the reviewer for this insightful suggestion. We have performed CellChat analysis on the dataset generated by Burl et al., 2022. The results of this analysis are presented in Supplementary Figure 2 and discussed in lines 191-204 of the revised manuscript. Importantly, the total number of significant connections between the cell types identified by Burl et al. also indicated that the adipocyte stem and progenitor cells (denoted as ASC1-3: adipose tissue stromal cell, Prolif/Diff: proliferating/differentiating cells in Burl et al.) are involved in the highest number of interactions with other cell types (Supplementary Figure 2c). This is consistent with the conclusion from our scRNA-seq dataset that adipocyte progenitors serve as the primary communication center, or "hub", in the brown adipose tissue microenvironment.

5) It is widely recognized that there are at least 2-3 ASC subtypes that express PDGFRA. The current data set does not distinguish among ASC subtypes, including adipogenic subtypes.

We appreciate the suggestion made by this reviewer and agree that it would be informative to explore the cellular crosstalk between various subsets of adipocyte progenitors. In our scRNA-seq dataset, unbiased cell clustering identified two subsets of Pdgfra-expressing adipocyte progenitors, namely Pdgfra+ APC 15 and Pdgfra+ APC 16. To compare the gene signatures of these subsets with those reported by other groups [1], we evaluated the expression of transcripts that are known to serve as markers of different subtypes (Response File_Table 1). Response File_Figure 1a depicts the expression of these markers in the two clusters of Pdgfra-expressing adipocyte progenitors. Our analysis revealed that these markers are not sufficient to differentiate the subsets of Pdgfra+ APCs in our BAT scRNA-seq dataset. It is important to note that the markers for APC subtypes were identified primarily using scRNA-seq analysis of inguinal and perigonadal white adipose tissue. Hence, these markers might not be effective in distinguishing the progenitors in BAT.

Despite this, we annotated the dataset to differentiate the two APC clusters and re-ran the CellChat analysis. Comparison of the interactions between cells in the Pdgfra+ APC 15 and Pdgfra+ APC 16 subsets revealed similar interaction patterns, suggesting that the two clusters of APCs interact with other cell types in a comparable manner (Response File_Figure 1b-c).

Response File_Table 1. Adipose stromal subpopulations identified by single-cell transcriptomics

Markers	APC subtype
Dpp4, Anxa2, Wnt2	Interstitial progenitor cells (Uncommitted progenitors)
Icam1, Pparg, Dlk1	Committed preadipocytes
Pdgfrb, Ly6c	Fibroinflammatory progenitors (FIPs)

Response File_Figure 1. (a) Gene expression of APC marker genes in the two clusters of Pdgfra+ APC in our dataset of BAT-SVF. (b) Circle plot showing the cell-to-cell communications involving the Pdgfra+ APC 15 as sender (top) and receiver (bottom). (c) Circle plot showing the cell-to-cell communications involving the Pdgfra+ APC 16 as sender (top) and receiver (bottom). The line width represents the number of ligand-receptor interactions between two cell types. The size of the circle represents the number of interactions in each cell type, line width represents the number of interactions between two cell types, and color distinguishes sender cell types.

6) The authors suggest there is important cellular communication between mature adipocytes and a variety of other cell types. However, captured BA are speculated to be newly differentiated cells since they did not float. Inspection of the published scRNA libraries indicates that the BA phenotype is limited to only one of the 4 2-day-cold libraries, and none of the 7-day libraries (reflected in supplemental figure 1A), raising the issue of reproducibility. Supplemental figure 1A should show individual values of all libraries. At this point it would be difficult to make claims concerning secreted angiogenic factors from mature cells.

We agree with the reviewer that the dataset used in this study is not suited for determining the crosstalk between mature adipocytes and other cells in the BAT microenvironment. We have amended the text to avoid misleading the readers.

The number of cells in the 'adipocytes' cluster in each library is listed below. In total, we detected 3 adipocytes in the TN group, 49 in the RT group, 948 in the cold2 group, and 27 in the cold7 group. Statistical analysis of the cell type frequencies showed a significant over-representation of the adipocytes cluster in mice housed in cold for 2 days (Wald test p-value 0.0005, Supplementary Figure 1a). Importantly, this is consistently observed for the samples in the Cold2 group and is not limited to only one sample. As discussed in the manuscript, this is likely caused by an increase in the number of differentiating adipocytes, which have lower lipid content and buoyancy than the fully mature adipocytes

TN group (GSM4875674: 3, GSM4875675: 0, GSM4875682: 0, GSM4875683: 0)

RT group (GSM4875684: 7, GSM4875685: 42, GSM4875676: 0, GSM4875677: 0)

Cold2 group (GSM4875679: 349, GSM4875686: 29, GSM4875687: 570, GSM4875678: excluded from all the analysis due to high background RNA)

Cold7 group (GSM4875680: 4, GSM4875681: 2, GSM4875688: 9, GSM4875689: 12)

7) It is a little surprising that B and T lymphocytes are identified as contributing to over 50% of the stromal population. Nonetheless, it is barely mentioned what type of cell chat is involved in this population. As a major cell type, there should be more evaluation of how they may contribute to cold-induced neogenesis.

Although our scRNA-seq data identified large populations of B and T lymphocytes, they are only engaged in a small number of interactions with other cell types in BAT (Figure 3a). Therefore, in the original manuscript, we chose to focus on other types of immune cells, including macrophages, NK cells, Neutrophils, ILC2s, and Tregs, as these cells are found to have more ligand-receptor crosstalk in the BAT microenvironment based on our CellChat predictions. Nevertheless, in response to this comment, we have added additional discussions about the potential crosstalk between adipocyte

progenitors and B-cells mediated by App-Cd74 interactions (lines 266-270 in the revised manuscript).

Reviewer #3 (Remarks to the Author):

In their manuscript Shamsi and colleagues apply a novel bioinformatic technique to single cell RNAseq data obtained from the BAT of mice kept at thermoneutrality and exposed to cold. Using the CellChat algorithm they find that changes in the ambient temperature elicit comprehensive alterations of cell-cell communications in murine BAT.

This is a very interesting manuscript and the findings are novel. However, several points should be revised and the conclusions drawn by the authors should be put into perspective.

Major points:

1. The authors analyzed the stromal-vascular fraction "SVF" of BAT samples from mice kept at different ambient temperatures for different amounts of time. In the manuscript the distinction between SVF and BAT is rarely made and conclusion are drawn for BAT as a whole. This is not true since mature adipocytes made up only a very small proportion of the analyzed cells. The reviewer is well aware of the fact that it is challenging to do scRNASeq of mature adipocytes but this is a major limitation of the paper and should be addressed appropriately.

We thank the reviewer for the suggestion and have modified the text to make a clearer distinction between BAT and BAT-SVF. Additionally, we have included a new section in the manuscript to discuss the limitation of this study (lines 390-416 in the revised manuscript).

2. I tried to figure out how many mice per treatment group were used but I couldn't find this data.

The single cell RNA-seq data were generated using four animals for each temperature condition. The method section and the Figure 1 legend have been amended to include the number of animals in each group.

3. Could the treatment of the tissue (mincing, digesting, straining,...) prior to isolation of the RNA influence the expression levels of the analyzed genes which are involved in cell-cell interactions?

We have amended the manuscript to include a section describing the limitation of this study including the potential effects of tissue processing and digestion on cellular transcriptome (lines 390-416 in the revised manuscript)

4. On which groups are the figures 3/4/5 a to c based: is this an analysis of all interventions groups (pooled) or only data from the animals kept at RT?

The data presented in Figures 3-5 are generated from the analyses of all groups. We have added these details in the figure legends 3-5.

5. cross-talk between adipogenic cells and vascular cells
There does not seem to be much of a difference in terms of cell chatting between the thermoneutral and RT groups (Figure 4 d). Additionally, I would expect more similarities between the cold2 and cold7 groups.

The heatmaps in Figure 4d present the top 50 interactions whose probability of interactions is changed with temperature. Among these top interactions, there is a higher proportion of those that are significantly regulated by 7 days of cold exposure. The heatmap also shows that the 2 days of cold exposure induce some similar changes, albeit to a lower extent. This suggests that the full activation of the pathways involved in cold-induced angiogenesis requires a more chronic cold challenge.

6. This seems to be the case for the interactions between adipogenic cells and Schwann cells (figure 5 d).

Similarly, the heatmaps in Figure 5d present the top 50 interactions whose probability of interactions is changed with temperature. Among these top interactions, there is a higher proportion of those that are significantly regulated by 7 days of cold exposure. The heatmap also shows that the 2 days of cold exposure induce some similar changes, albeit to a lower extent. This suggests that the full activation of the pathways involved in cold-induced innervation requires a more chronic cold challenge.

Discussion / Conclusion

In general these data are interesting but they are largely descriptive in nature and do only rely on the transcriptome. As stated by the authors the software used makes predictions on cell-cell interactions and does not give unequivocal proof of the cell communications. The data can very well serve as a starting point to generate new hypotheses about BAT differentiation and regulation which need to be tested in the future. These limitations should be discussed appropriately.

We thank the reviewer for this suggestion. A paragraph discussing this limitation has been included in the new limitation section (lines 390-416 of the revised manuscript).

line 371: "Collectively, our integrative analysis of intercellular crosstalk in BAT provides a holistic understanding of the mechanisms that control thermogenesis."

- This is a very bold statement given the rather descriptive nature of the findings in this paper. The following sentence is much closer to reality: it is "a valuable platform for future investigations"

We agree with the reviewer's comment and have modified the text accordingly.

Minor points:

Figure 2: the black dots should mark $p_{\text{adjusted}} < 0.05$ and not ≤ 0.05

We thank the reviewer for pointing out the error. This has been corrected in the revised Figure 2.

Figure 3/4/5: Why did the authors use color and size to display the number of cell-cell interactions within the graphs A and C. This seems to be redundant.

The graphs in Figure 3-5 have been modified to eliminate the redundancy in the representation.

REVIEWERS' COMMENTS:

Reviewer #2 (Remarks to the Author):

The authors have satisfactorily addressed issue raised in first review.

Reviewer #3 (Remarks to the Author):

The authors have answered my questions / remarks satisfactorily. I recommend to accept this manuscript for publication.